

**Climatic characteristics of the Jianghuai cyclone and its linkage with**
**precipitation during the Meiyu period from 1961 to 2020**
Ran Zhu[1], Lei Chen[1,2]
[1]Department of Atmospheric Science, School of Environmental Studies, China University of
Geosciences, Wuhan, 430074, China
[2]Centre for Severe Weather and Climate and Hydro-Geological Hazards, Wuhan, 430074, China
Correspondence to: Lei Chen (leichen@cug.edu.cn)
**Abstract.** This study examines the climatic characteristics of 202 Jianghuai cyclones
and their linkage with precipitation during the Meiyu period from 1961 to 2020. The
results show that cyclones mainly originate from eastern Hubei Province and south-
central Anhui Province. The frequency of Jianghuai cyclone occurrences shows an
increasing trend in 1965-1970, 1990-2000, and after 2020. A decreasing trend is
observed for 1970-1990 and 2000-2010. There is a positive correlation coefficient of
0.769 between the frequency of cyclone activity and precipitation in the Meiyu period.
The percentage of precipitation affected by cyclone activities can reach up to 47%. The
anomalous increase in precipitation caused by cyclones above 27°N can reach a
maximum of 7 mm/day. In contrast, precipitation is decreased in southern China
because of the strengthening positive geopotential height in the Western Pacific
subtropical high (WPSH). Precursor negative geopotential height anomalies for these
cyclones emerge over the Mongolian region. The abnormal signal of the negative
geopotential height can be traced to day -4 at the 500 hPa level. The WPSH and
southwest jet are the dominant factors causing abnormal precipitation during Jianghuai
cyclones. The significant water vapor convergence anomalies in the middle and lower
reaches of the Yangtze River and the southwestward water vapor transport anomalies
provide sufficient water vapor for cyclone development.



## 1. Introduction

The Meiyu front is one of the important weather systems affecting summer
precipitation in the middle and lower reaches of the Yangtze River (Pang et al., 2013;
Wang et al., 2014; Zhou et al., 2022; Tao et al., 1979). From mid-June to early July, the
east of Yichang, Hubei Province, has continuous rains and short sunshine. These
conditions are accompanied by heavy rainfall, strong wind and other weather
phenomena in these areas during the Meiyu period (Ding. 1992,1994; Zhao et al., 2021;
Zhou et al., 2016). In China, the mean annual precipitation during the Meiyu period in
the Jianghuai River Basin can reach 300 mm, accounting for 30%-40% of the mean
annual total precipitation, and even up to 500 mm or more in the extreme Meiyu period
(Liu et al., 2020). Historically, most of the summer flood disasters are caused by
precipitation anomalies in the Meiyu period. Some scholars have studied and analyzed
the representative floods of 1996, 1998, 2016 and 2020 (Bao et al., 2021; Su et al., 2021;
Zhao et al., 2018; Zhong et al., 2023). These floors, caused by the Meiyu front, had
adverse effects on people's safety, lives and property (Yan et al., 2021). Scholars in
China have divided rainstorms caused by Meiyu fronts into three types (Zhang et al.,
2014). The first type is the β mesoscale convective rainstorm on the Meiyu front. This
type of rainstorm has a range of less than 300 km with strong intensity and a fast
formation process (He et al., 2007). It is difficult to forecast before 12 hours and can be
detected only by using radar to make a proximity forecast (Zhang et al., 2002). The
second type is the persistent rainstorm located in front of the high-altitude low-pressure
tank in the western part of the Meiyu front. It is characterized by a long duration of
approximately 5 days but appears less frequently, mainly in western Hubei and western
Hunan and Sichuan (Cai et al., 2021; Wu et al., 2020;). The last type is the rainstorm
caused by the Jianghuai cyclone located east of the origin of the Meiyu. The Jianghuai
cyclones are affected by the thermal conditions of the sea and land and likely occur in
the eastern part of the Meiyu front (Wang et al., 2016). The positive vorticity advection
in front of the high-altitude trough and the warm advection in front of the front promote
the eastward movement and development of the cyclone (Shen et al., 2019; Zhang et





al., 2016). During the development of the cyclone, the lower levels are dominated by
the southwest warm and humid airflow, and the high levels are mainly affected by dry
and cold air (Zhao et al., 2008). This type of rainstorm has a large range, high intensity
and long duration of precipitation (Wang et al., 2012; Xu et al., 2011).

Scholars' studies on Jianghuai cyclones during the Meiyu period were initially

based on individual case analysis. Xu et al. (2013) studied a cyclone process in 2011
and found that the cyclone process lasted up to 36 h. The cyclone rainstorm was
distributed on the south side of the cyclone. Heavy precipitation during the whole
cyclone mainly occurred in the lower reaches of the Yangtze River. Wu et al. (2020)
studied 2 different cyclone rainstorm processes. They found that rainfall is directly
proportional to cyclone intensity. There is a strong convergence center of water vapor
flux during cyclone development. Zhou et al. (2020) found that a tornado was generated
from the cyclone occlusion stage on July 22. The tornado was under the influence of a
strong and fast Jianghuai cyclone and produced heavy precipitation accompanied by
thunderstorm phenomena. With the improvement of cyclone identification methods and
reconstruction of reanalysis data, statistical studies of cyclones have been further
developed (Simmonds et al., 2000; Wernli et al., 2006). Yang et al. (2010) modeled the
rainstorm process in the lower reaches of the Yangtze River from 1998 to 2005. The
cyclones accounted for 62.5% of the rainstorm cases, and more than 70% of the
cyclones could develop and produce rainstorms. The Jianghuai cyclone located in the
lower reaches of the Yangtze River generally exists in the lower troposphere at 700 hPa.
The horizontal scale is within 400 km, and the life period on land is generally less than
48 h. Wang et al. (2015) found that the number of cyclones was lower and their intensity
was weak in the 1980s and 1990s. In the early 2000s, cyclones were more frequent, and
their intensity increased. After 2010, there was again a decreasing trend. Zhang et al.
(2018) divided 60 cases of extreme precipitation in the middle reaches of the Yangtze
River from 2008 to 2015 into five types. Among them, the extreme precipitation of the
Jianghuai cyclone type accounted for 30%. The stable and maintained Western Pacific
subtropical high (WPSH) system is one of the important reasons for the strong



precipitation produced by cyclones. Because of the weak cold air force, the intensity of
the Jianghuai cyclone is weaker than that in spring (Zhou et al., 2017). The daily
analysis of the Jianghuai cyclones in the Meiyu period is easy to ignore. All these
studies indicate that the Jianghuai cyclone is an important weather system that causes
heavy rainfall during the Meiyu period in the middle and lower reaches of the Yangtze
River (Wu et al., 2021; Zhang et al., 2018; Zhu et al., 1998).

Research on the climatic characteristics and precipitation effects of Jianghuai

cyclones during the Meiyu period in the past 60 years has not yielded clear results. In
this study, the relative vorticity method is used to objectively identify and track
cyclones based on reanalysis data provided by ERA5. The climatological characteristics
of the Jianghuai cyclones during this period are studied. We analyze the correlation
between Jianghuai cyclone activity and precipitation. This study provides a reference
for the long-term and short-term forecasting of precipitation in the Meiyu period.

The remainder of the present paper is organized as follows. Section 2 of this paper

presents the dataset and analytical methods. In Section 3, we show the climatology
composite of the cyclone tracks, genesis locations, intensity, lifetime and so on. There
is a positive correlation between the frequency of cyclonic activity and precipitation in
the Meiyu period. The relationship between them is studied by means of the
geopotential height anomaly and water vapor flux anomaly. Section 4 provides the main
discussion and findings of this study.
**2.  Data and methods**
**2.1 Data**
The time span of all the data is 60 years from 1961 to 2020, and the study area is
located at 108°E-123°E, 27°N-34°N. We use the ERA5 relative vorticity hourly data
(850 hPa) released by the European Centre for Medium Range Weather Forecasts
(ECMWF) for Jianghuai cyclone identification and tracking. The spatial resolution of
the data is 0.25°×0.25°, and the temporal resolution is 6 h. Every 6 h was defined as a
step. The data of geopotential height, wind field, and specific humidity are daily data



processed from ERA5 hourly data with a spatial resolution of 0.25°×0.25° (Hersbach
et al., 2018). The geopotential height and wind field data include pressure levels of
approximately 500 hPa, 700 hPa and 850 hPa. The specific humidity data include
pressure levels of approximately 500 hPa, 700 hPa and 850 hPa. The precipitation data
are from the CN05.1 grid point observation dataset compiled by the National
Meteorological Information Center with a resolution of 0.25°×0.25°.
**2.2 Methods**
The objective identification and tracking method for cyclones used in this paper is
the vorticity tracking method proposed by Hodges (1994, 1995). The first step is to use
the relative vorticity field at the 850 hPa pressure level corresponding to every moment
of the cyclone to determine the range of each cyclone. The second step is to find the
feature points. In the process of finding the feature points, the extreme point and the
centroid point are the alternatives. Corresponding to the global relative vorticity grid
data of each time point, several feature points can be found, and each point represents
a cyclone. The third step is to match the track of each cyclone under the given time
span. In Hodges (1994), the assumed data used are defined on a rectangular grid, and
each time step is initially processed to identify the maximum or minimum value of the
"object" on the positioning grid. The tracking method is feasible on high-resolution
grids, but on low-resolution grids, the feature points may not be sufficient to produce
smooth trajectories, so the smoothness of the tracking algorithm is effectively limited.
Hodges (1995) proposed tracking feature points on the unit sphere, which would
become the feature point matching problem of grid data for adjacent time points in
cyclone tracking. If the algorithm is reasonable, there is no "discontinuity" mutation in
the final arriving cyclone track, and the track is more accurate.
In addition to the relative vorticity method of tracking proposed by Hodges,
different methods of cyclone identification have also been proposed by other scholars.
Lu (2017) improved the extratropical cyclone identification and tracking method
involving the nine-point pressure minimum. Jiang et al. (2020) proposed an algorithm



for identifying extratropical cyclones on the basis of gridded data. This algorithm is
named the eight-section slope detection method.

Based on many different methods of objective cyclone identification, we chose the

relative vorticity tracking method. The relative vorticity tracking method can detect low
vortex systems earlier and track cyclones for a longer period of time with better stability.
When the closed pressure levels are not visible on the satellite map, the vorticity
tracking method can still continue to track the cyclone, improving the accuracy of
cyclone track data.
**3. Results**
**3.1 Climatic characteristics of the Jianghuai cyclone during the Meiyu period**

A total of 202 Jianghuai cyclones existed during the Meiyu period from 1961 to

2020. The range of cyclone genesis locations defined by the Jiangsu Meteorological
Administration (2017) and the characteristics of the relative vorticity tracking method
were used. We adjust the genesis location and remove the cyclones that are generated
at sea and have no effect on land precipitation (108°E-123°E, 27°N-34°N). Figure 1a
shows the distribution of Jianghuai cyclone tracks. The brown dots represent the genesis
locations of the first occurrence of the Jianghuai cyclone. The yellow lines indicate the
tracks of the cyclones. As shown in the figure, the tracks of the cyclone are mainly
eastward and northeast. These two kinds of tracks are related to the upper-level guide
airflow of 500~700 hPa (Wei et al., 2013). The northeast track is mainly due to the
southwest warm and moist air on the edge of the WPSH. The east track is mainly related
to the location of the WPSH. Figure 1b shows the frequency of cyclone genesis
locations during the Meiyu period from 1961 to 2020. The genesis locations of cyclones
are mainly located in the middle and lower reaches of the Yangtze River and the Huaihe
River basin, with an east–west band distribution (Wang et al., 2015; Wu et al., 2020).
The frequency of occurrence is higher in the region of the Hubei and Chongqing
junction, eastern Hubei, northern Jiangxi, south-central Anhui, Jiangsu and Zhejiang.
Research has found that the genesis locations of cyclones are closely related to the



landform (Xu 2021; Zhang et al., 2012).

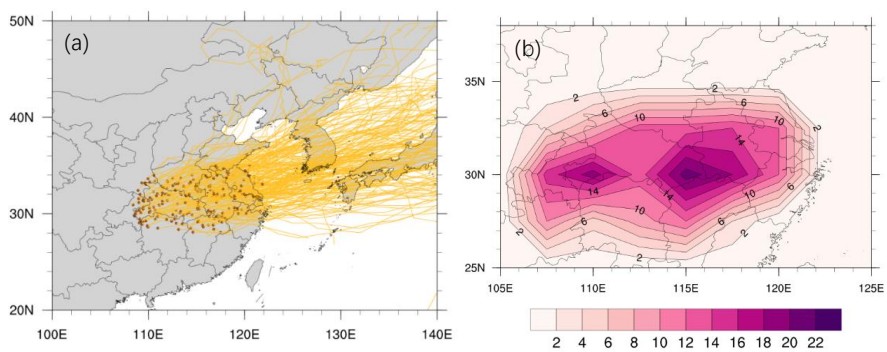

Fig 1. Distribution of the cyclone genesis locations and the cyclone tracks during the
1961-2020 Meiyu period (a). (The brown dots represent the genesis locations. The
yellow lines indicate the tracks). The frequency of genesis locations for the selected
cyclones during the Meiyu period from 1961 to 2020 (b).

To examine the climatological characteristics of Jianghuai cyclones over 60 years,

we focus on the intensity, radius, and lifetime of cyclones on land. The intensity of the
Jianghuai cyclone is defined as the relative vorticity intensity of the 850 hPa cyclone
center. The larger the relative vorticity intensity is, the stronger the cyclone intensity is.
Figure 2a shows that among the 202 selected cyclones, the intensity of the cyclone
center mainly ranges from $0\times10^{-5}$ s$^{-1}$ to $7\times10^{-5}$ s$^{-1}$. When the intensity of the cyclone
center is less than $3\times10^{-5}$ s$^{-1}$, the number of cyclones increases with increasing intensity;
when it is larger than $3\times10^{-5}$ s$^{-1}$, the number of cyclones decreases with weakening
intensity. The number of cyclones in the range of $2\times10^{-5}$ s$^{-1}$ to $3\times10^{-5}$ s$^{-1}$ has the largest
proportion, accounting for 36% of the total number of cyclones. A total of 180 cyclones
are in the range of $1\times10^{-5}$ s$^{-1}$ to $5\times10^{-5}$ s$^{-1}$ in intensity, accounting for 89%. Figure 2b
shows the relationship between the radius of cyclones and the number of cyclones. Most
of the cyclones have an average radius between 300 and 800 km, accounting for 96%
of the total number. The number of cyclones with radii between 500 and 600 km is the
largest, accounting for 35%. Figure 2c shows the relationship between the time of
cyclones affecting precipitation on land and the number of cyclones. Most of the





cyclones affect precipitation on land for 1-3 days, and only one cyclone affects
precipitation on land for more than 3 days. The number of cyclones that affected
precipitation on land within 2 days was 186, accounting for 92% of the total number.
The intensity of a cyclone is one of the factors affecting its precipitation and impact
range during the Meiyu period (Zhao et al., 2010). Figure 3a shows a positive
correlation between the maximum intensity and the maximum radius of cyclone
development. The stronger the intensity of a cyclone is, the larger its radius. Therefore,
the horizontal scale of most strong cyclones is larger than that of weak cyclones, the
precipitation is greater, and the precipitation range is larger. From the distribution of the
time lag between the maximum intensity and the radius of the cyclone shown in Figure
3b, the number of cyclones that reach both at the same time accounts for 45% of the
total number of cyclones. Of the remaining Jianghuai cyclones, more reach the
maximum intensity first and then continue to develop to the maximum horizontal scale.

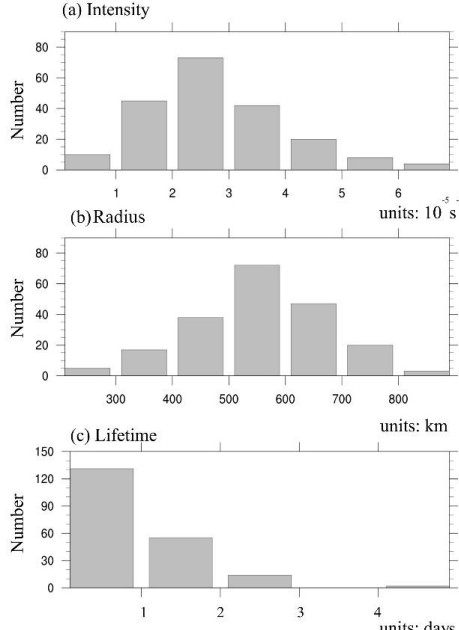

Fig 2. Distributions of the number of selected cyclones versus their (a) intensities (units:
$10^{-5}$ s$^{-1}$), (b) radii (units: km), and (c) lifetimes (units: days).





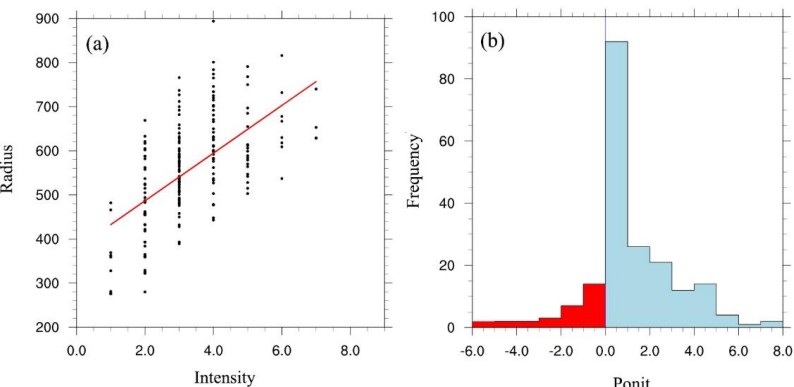

Fig 3. Correlation between maximum intensity (units: $10^{-5}$ s$^{-1}$) and maximum radius (units: km) (a) and their time lag during the development of the Jianghuai cyclone in the Meiyu period (b).

The frequency of Jianghuai cyclone occurrence is characterized by multiperiod variation (Figure 4). The shaded area in the figure indicates that the 95% confidence interval is passed. Strong 2–4-year quasiperiodic variation is observed for 1980-1990 and 1990-2000. After 2000, the quasiperiodic change in cyclones is approximately 4-5 years. This change period corresponds to the period of abnormal change in Meiyu. Chen et al. (2019) pointed out that 3~4 years of quasiperiodic change is the main component of abnormal changes in Meiyu when studying the quasiperiodic change in Meiyu. This quasiperiodic variation component is mainly influenced by the out-of-ocean forcing of the Indian Ocean dipole (IOP), which changes from the ENSO in the previous winter to late spring and early summer with seasonal changes (Liang et al., 2018). During the positive phase of the IOP, the strong warming of the Indian Ocean triggers a strong Indian monsoon. This leads to a strengthening of the WPSH and an increase in precipitation in southern China. The southwestern rapids, which are enhanced by the positive IOP, also provide sufficient water vapor and warm advection to generate favorable conditions for the development of the Jianghuai cyclone.



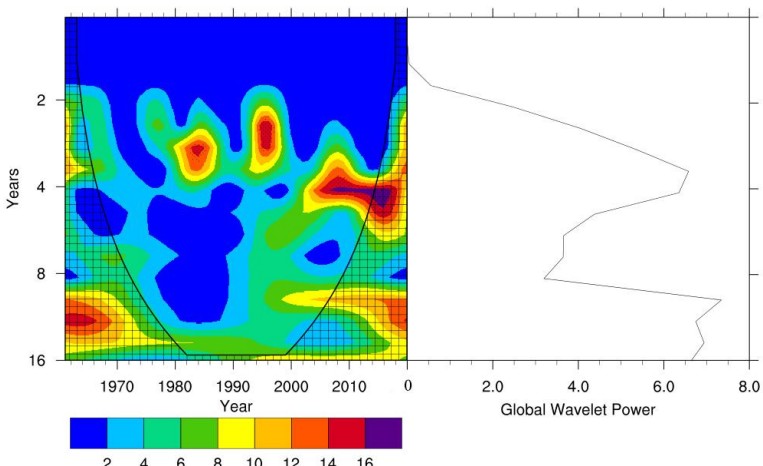

Fig 4. Periodic wavelet analysis diagram of Jianghuai cyclones during the Meiyu period
from 1961 to 2020 (shadow indicates passing the 95% confidence interval).

Jianghuai cyclones are not only characterized by multiperiod variability but also

have significant interdecadal variability. Figure 5 shows the activity frequency anomaly
and 5-year sliding average of cyclones during the Meiyu period from 1961 to 2020. The
frequency of cyclone activity was the highest in 1996 and the lowest in 1961 and 2009.
In the long term, the frequency of cyclone activity in the middle and lower reaches of
the Yangtze River increased in 1965-1970, in 1990-2000 and after 2010. It decreased
in 1970-1990 and 2000-2010. The interdecadal variability trend of Jianghuai cyclones
is similar to the interdecadal variability trend of precipitation during the Meiyu period
(Chen et al., 2019).

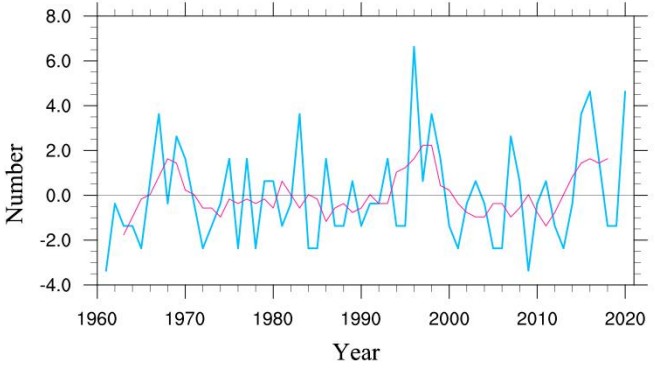



Fig 5. Frequency anomaly and 5-year sliding average of cyclones. The blue line
shows the anomalies in the number of cyclones, and the pink line shows the 5-year
sliding average of the anomalies.

## 3.2 Linkage between cyclone activity and concurrent rainfall in the middle and lower reaches of the Yangtze River.

The Jianghuai cyclones are mainly active in the middle and lower reaches of the
Yangtze River (Huang et al., 2019; Li et al., 2002). Under the influence of the
strengthening westward extension of the WPSH during the Meiyu period, the Jianghuai
cyclones are restricted from entering the sea to some extent (Qin et al., 2015; Wu et al.,
2020). They form rainstorms and gales in the middle and lower reaches of the Yangtze
River and the coastal areas. A large part of the precipitation in the Meiyu period comes
from cyclone precipitation (Zhang et al., 2018). The intensity of Meiyu is usually
expressed by the Meiyu intensity index. The intensity of precipitation is affected not
only by precipitation but also by the number of precipitation days in the Meiyu period.
Both jointly determine the intensity of Meiyu in that year. The time-series plots of the
number of cyclones related to precipitation and the intensity index during the Meiyu
period from 1961 to 2020 are given in Figure 6a and 6b. We found that the number of
cyclones has a positive correlation coefficient of 0.769 with precipitation in the Meiyu
period (passing the 99% confidence interval). The number of cyclones was also
positively correlated with the Meiyu intensity index, with a correlation index of 0.760
(passing the 99% confidence interval). The frequency of Jianghuai cyclone activity in
years with a strong Meiyu index is high; the frequency of Jianghuai cyclone activity in
years with a weak Meiyu index is low.

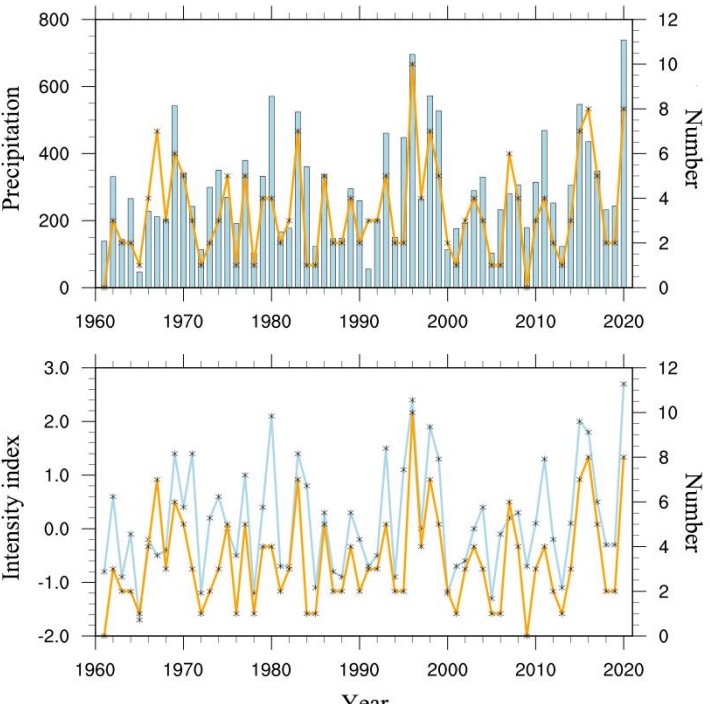

Fig 6. (a) Changes in precipitation (blue bar chart) (unit: mm/day) and the number of cyclones (orange line); (b) intensity index (blue line) and the number of cyclones (orange line) in the Meiyu period from 1961 to 2020.

Figure 7a shows the spatial distribution of annual average precipitation during the Meiyu period from 1961 to 2020. The areas with large precipitation values in the middle and lower reaches of the Yangtze River are mainly located in the Dabie Mountains of Anhui Province, the northern part of Jiangxi Province, the eastern part of Hubei Province and the western part of Hubei Province. The maximum annual average precipitation during the Meiyu period in southern Anhui can even exceed 480 mm. The occurrence of large precipitation areas during the Meiyu period is closely related to the topography of the region (Wu et al., 2023).

If precipitation and Jianghuai cyclone activity existed on the same day during the Meiyu period, we defined that day as a Jianghuai cyclone precipitation day. The remaining days in the Meiyu period were treated as non-Jianghuai cyclone precipitation





days. Figure 7b shows the spatial distribution of the proportion of cyclone precipitation
relative to total precipitation during the Meiyu period. As shown in the figure, the main
areas affected by cyclone precipitation are the middle and lower reaches of the Yangtze
River. The Huaihe River basin in northern Anhui Province is the most affected area.
The cyclone precipitation in the Huaihe River basin accounts for more than 47% of the
total precipitation during the Meiyu period, while the cyclone-influenced precipitation
in other areas accounts for more than 35% of the total precipitation. In general, the
degree of cyclone-influenced precipitation in the middle and lower reaches of the
Yangtze River shows an east–west band distribution and a gradual decrease from
coastal to inland areas. This indicates that the distribution of the large-value area and
the characteristics of the band distribution are related to the northeast and eastward
tracks of the Jianghuai cyclone. Its precipitation capacity gradually increases with the
development of cyclone movement.

Figure 7c shows the spatial distribution of the daily mean precipitation anomaly

of the Jianghuai cyclone (the shaded part indicates that the 95% confidence interval is
passed). When the Jianghuai cyclone is active, the middle and lower reaches of the
Yangtze River to the east of 108°E show an abnormal increase in precipitation. However,
Fujian, Guangdong and other places show an abnormal decrease. Among them, the
maximum value of abnormally increased precipitation can exceed 7 mm/day in areas
such as southern Anhui, eastern Hubei and northern Jiangxi. The large-value areas of
precipitation anomalies are consistent with the large-value areas of cyclone occurrence
frequency sources. It is inferred that the spatial distribution of precipitation anomalies
has a connection with the distribution of cyclone genesis locations. This phenomenon
of increasing and decreasing precipitation anomalies is bounded by approximately
27°N and distributed north–south in the form of dipoles.





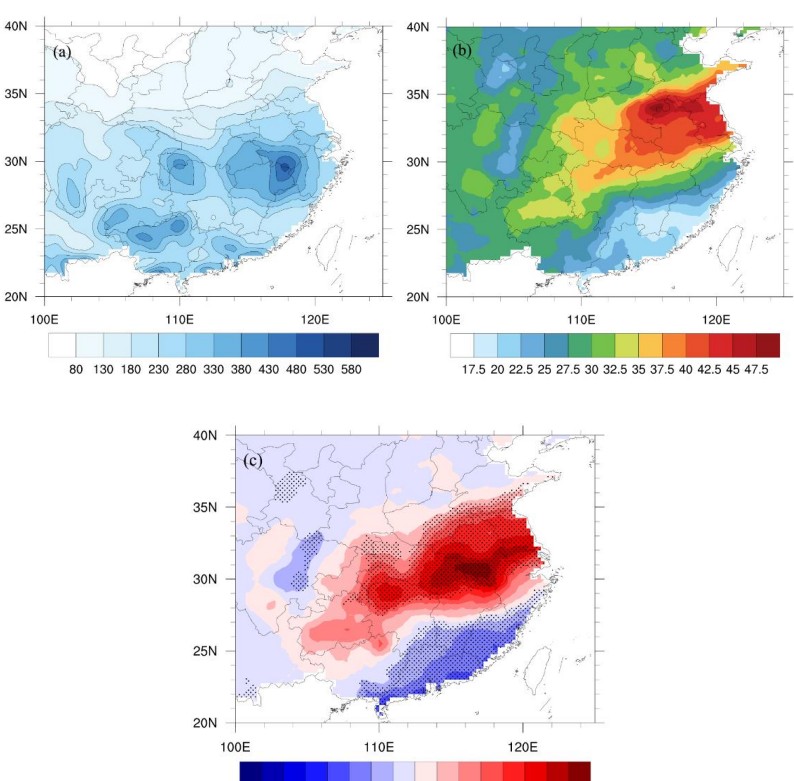

Fig 7. (a) Annual mean precipitation during the Meiyu period from 1961 to 2020 (unit: mm/year); (b) proportion of Jianghuai cyclone precipitation relative to total precipitation during the Meiyu period (unit: %); (c) daily mean precipitation anomaly of the Jianghuai cyclone during the Meiyu period (unit: mm/day) (shadow indicates passing the 95% confidence interval).

Figure 8 shows the evolution of composite geopotential height and horizontal wind anomalies for three different levels of Jianghuai cyclones from day -4 to +2 during the Meiyu period. Day 0 is the day on which the cyclone first appears in the specified area. Most areas of the lower and middle troposphere (700 hPa, 850 hPa) in the middle and lower Yangtze River on day 0 are covered by significant negative geopotential height anomalies with peak magnitudes greater than -11 gpm. There is a significant positive geopotential height anomaly with a peak magnitude of over 13 gpm on the southeast





side of the negative geopotential height anomaly. These anomalies form meridional
dipole structures in the middle and lower troposphere geopotential height field. The
southwest wind anomaly is significant in the middle and lower reaches of the Yangtze
River. The south of Anhui Province and the north of Jiangxi Province are between the
positive geopotential height anomaly and negative geopotential height anomaly. The
existence of these anomalies indicates the enhancement of southwest rapids and the
strengthening of the WPSH. The negative geopotential height anomalies at 500 hPa
height on day 0 are mainly in Mongolia, Shanxi and other places. Strong southwest
wind anomalies exist between the positive and negative geopotential height anomalies.
The negative geopotential height anomalies in the Mongolian region exceed -7 gpm.

The negative geopotential height anomalies on all three isobaric surfaces can be

traced back to Mongolia, Inner Mongolia and part of Northeast China on day -2.
Negative geopotential height anomalies at 500 hPa can be traced to day -4. On day -4,
significant southwestern wind anomalies exist in southwestern Hunan at 700 hPa and
850 hPa. Significant northwest wind anomalies exist in the Yellow River basin of China
at 500 hPa. By day -2, the negative geopotential height anomalies in Mongolia, Inner
Mongolia and some northeastern areas are enhanced for all three isobars. The positive
geopotential height anomalies of the WPSH are enhanced and extend northward to the
southern part of the middle and lower reaches of the Yangtze River. There are
significant southwest wind anomalies at the three isobaric surfaces in the south of the
middle and lower reaches of the Yangtze River, while there are significant northwest
wind anomalies at 500 hPa in the north of Anhui Province and Jiangsu Province. The
negative geopotential height anomalies on the three isobaric surfaces move eastward
with the formation and development of Jianghuai cyclones. On day +2, the lower
reaches of the Yangtze River are mainly affected by the combined action of anomalous
southwest winds and northwest winds. The positive geopotential height anomaly of the
WPSH is weakened.

Therefore, the abnormal precipitation caused by the Jianghuai cyclone mainly

comes from the abnormal southwest winds and the strengthening of the WPSH



(Rodwell et al., 1996; Sardeshmukh et al., 1998). The enhanced negative geopotential
anomaly over Mongolia provides cold and dry air brought by the westerly jet for
cyclone development. The enhanced southwest jet provides sufficient warm and moist
air for the formation of cyclones and promotes the eastward migration of cyclones after
formation. The increasing frequency of cyclones over the Yangtze River and Huaihe
River leads to the abnormal increase in precipitation in the middle and lower reaches of
the Yangtze River during the Meiyu period. However, due to the strengthening of the
WPSH, the southern part of China is controlled by the abnormal positive geopotential
height, and the precipitation decreases (Liu et al., 2020).

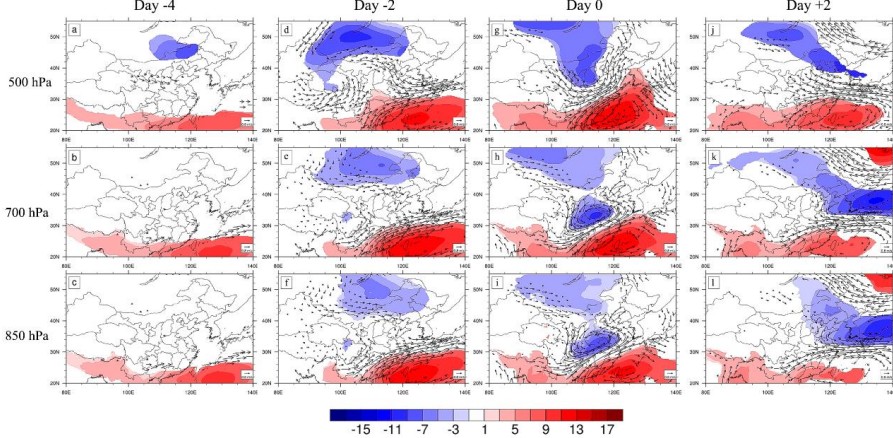

Fig 8. Evolution of composite geopotential height anomalies (shading; units: gpm) and
horizontal wind anomalies (units: m/s) on the 850 hPa, 700 hPa, and 500 hPa isobaric
surfaces for day −4 (a–c), day −2 (d–f), day 0 (g–i) and day +2 (j–l) for the 202 selected
Jianghuai cyclones. Shading indicates that composite geopotential height anomalies are
significant at the 95% confidence level based on a T test. Vectors are plotted if wind
anomalies are significant at the 95% confidence level based on a T test in at least one
direction.
Figure 9 shows the climatic distribution of water vapor flux and water vapor flux
divergence at a pressure level of 850 hPa during the Meiyu period. The water vapor
involved in the precipitation process of the Jianghuai cyclone during the Meiyu period



mainly comes from the water vapor brought by the southwest jet of the summer monsoon in the low-latitude area. During Jianghuai cyclone development, the middle and lower reaches of the Yangtze River are mostly in the water vapor convergence area, which is conducive to the generation of precipitation (Chen et al., 2020).

Figure 10 shows the distribution of water vapor flux anomalies and water vapor flux divergence anomalies at the pressure level of 850 hPa during the Jianghuai cyclone from day -2 to day +2. The color field and wind vector arrows in the figure both passed the 95% significance test. On day -2, a significant water vapor convergence anomaly and water vapor transport in the southwest direction appear in southern Anhui Province. The anomalies of water vapor flux and water vapor flux dispersion are mainly concentrated on day 0. There is significant anomalous water vapor convergence up to -1 $g \cdot cm^{-2} \cdot hPa^{-1}$ in eastern Hubei Province, Anhui Province and Jiangsu Province on day 0. Anomalous water vapor dispersion exists in the southern part of the middle and lower reaches of the Yangtze River and some areas in southern China. On day +2, with the development of the cyclone's eastward movement, only the southern part of Jiangsu Province and the northern part of Zhejiang Province have abnormal water vapor flux in the eastward direction. The precipitation in the area begins to gradually weaken at this time.

From day -2 to day 0, the abnormal water vapor flux and water vapor flux divergence configuration make the warm and wet air in the low-latitude area transport to the middle and lower reaches of the Yangtze River. The abnormal water vapor flux has a negative value, water vapor convergence occurs, local water vapor volume increases, and finally, the precipitation in the region increases. In contrast, the anomaly of water vapor flux in southern Guangdong and other regions is divergent. This leads to a decrease in local water vapor volume and precipitation in this region. These results indicate that the variations in water vapor flux and divergence related to cyclones are mainly from warm and wet air transported from low latitudes to the middle and lower reaches of the Yangtze River. Therefore, there is a positive correlation between cyclone activity and precipitation in the middle and lower reaches of the Yangtze River.

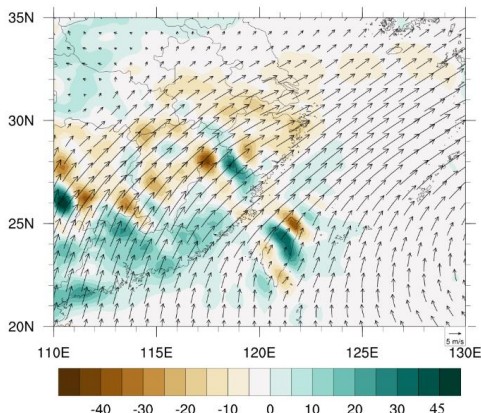

Fig 9. Distribution of 850 hPa daily mean water vapor flux (unit: $g \cdot cm^{-2} \cdot hPa^{-1}$) and water vapor flux divergence (unit: $10^{-8}\ g \cdot cm^{-2} \cdot hPa^{-1} \cdot s^{-1}$) of cyclones over the Yangtze and Huaihe rivers during 1961-2020 (color diagram shows water vapor flux divergence, and vector diagram shows water vapor flux). The colored region passed the 95% confidence interval according to a T test. If the vapor flux anomaly is significant at the 95% confidence level for the T test in at least one direction, the vector is plotted.

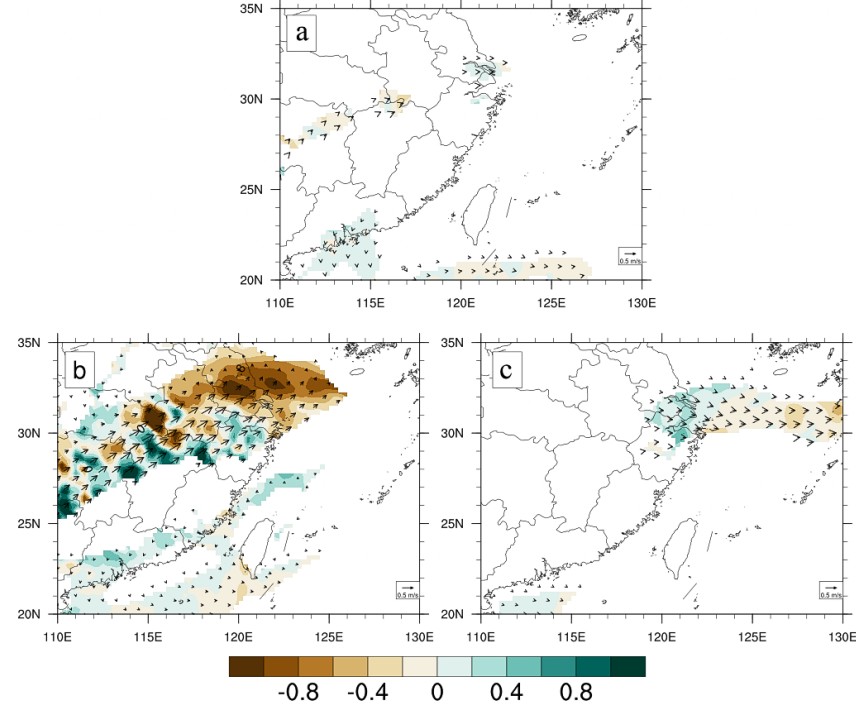



Fig 10. Distribution of the 850 hPa daily mean water vapor flux anomaly (unit: $g \cdot cm^{-2} \cdot hPa^{-1}$) and water vapor flux divergence anomaly (unit: $10^{-8}$ $g \cdot cm^{-2} \cdot hPa^{-1} \cdot s^{-1}$) of cyclones over the Yangtze and Huaihe rivers during 1961-2020 (color diagram shows water vapor flux divergence, and vector diagram shows water vapor flux). The colored region passed the 95% confidence interval according to a T test. If the vapor flux anomaly is significant at the 95% confidence level for the T test in at least one direction (zonal or meridian), the vector is plotted.

## 4. Summary and discussion

Based on ERA5 reanalysis of sea level pressure data and using the relative vorticity method to identify and track cyclones, we have examined the impacts of the climatological characteristics of Jianghuai cyclones. The linkages between cyclone activity and precipitation in the middle and lower reaches of the Yangtze River during the Meiyu period are also analyzed.

During the Meiyu period, Jianghuai cyclones are mainly generated at the junction of western Hubei and Chongqing Municipality, eastern Hubei Province, northern Jiangxi Province, central and southern Anhui Province, and Jiangsu and Zhejiang provinces. These cyclones develop and move to the sea in the east or northeast direction. There is a positive correlation between the maximum intensity and maximum radius of Jianghuai cyclones. The higher the cyclone intensity is, the larger the radius will be. Its occurrence frequency not only has the characteristics of multicycle variation but also has obvious interdecadal variation, which has a good correspondence with the periodic and interdecadal variation in precipitation in the Meiyu period.

There is a positive correlation between the frequency of cyclone activity and precipitation in the Meiyu period. The frequency of Jianghuai cyclone activity is high in the years with strong Meiyu rainfall and low in the years with weak Meiyu rainfall. The percentage of precipitation affected by Jianghuai cyclone activity in the middle and lower reaches of the Yangtze River can reach up to 47%. The spatial distribution is in the shape of an east–west belt, and the degree of influence gradually decreases from the



coast to the interior. When the Jianghuai cyclone is active, the precipitation increases
abnormally in the middle and lower reaches of the Yangtze River east of 108°E.
Precipitation decreases abnormally in Fujian Province and Guangdong Province. The
spatial distribution of precipitation anomalies is related to the genesis locations of
cyclone frequency, and the positive and negative anomalies are distributed north–south
in the form of dipoles based on the latitude line at approximately 27°N as the boundary.
The geopotential height anomaly field and the horizontal wind vector anomaly
field of the Jianghuai cyclones during the Meiyu period are synthesized and analyzed.
There is an enhanced positive geopotential height anomaly of the WPSH during cyclone
activity. The negative geopotential altitude anomaly of Mongolia and the abnormal
southwest jet are enhanced. All of these factors lead to an increase in precipitation in
the middle and lower reaches of the Yangtze River. The abnormal leading signal of the
negative geopotential height in Mongolia can be traced to day -2 of the cyclone activity,
and the signal can be traced to day -4 at 500 hPa. From day -2 to day 0 of cyclone
activity, the abnormal distribution of water vapor flux and water vapor flux divergence
cause the warm and wet air at the low latitudes to be transported to the middle and lower
reaches of the Yangtze River. They promote the generation and development of
cyclones and increase precipitation in the middle and lower reaches of the Yangtze
River. It is worth noting that different intensities of Jianghuai cyclones in the middle
and lower reaches of the Yangtze River may have different impacts on precipitation.
The specific mechanism by which the southwest jet affects cyclones during the Meiyu
period is not clear enough. These problems need further analysis and research.
**Competing interests**
The contact author has declared that none of the authors has any competing interests.



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
