# Peer review of "precipitation during the Meiyu period from 1961 to 2020"

_EGUsphere, 2023_

## Author Comment (AC1)

* * *
**Responses to reviewers' comments**

**In this response file,** the text in black **shows the comments from reviewers and editor,** **while** the text in blue **is our replies.**
* * *
This manuscript analyses Jianghuai cyclones during the Meiyu period using ERA5 reanalysis data. It explores cyclone characteristics, genesis locations, and tracks, establishing a positive correlation between intensity, radius, and interdecadal variations. The study emphasizes the link between cyclone activity and Yangtze River precipitation. Spatially, abnormal precipitation patterns are identified, tracing the evolution of geopotential height anomalies and water vapor flux.

The results presented in this paper are both interesting and significant, as the study delves into understanding the complex dynamics of Jianghuai cyclones and their impact on regional precipitation. While the methodology employed by the authors appears sound, there are a few areas that require revision. For example, I think it is advisable to provide more details on the parameters and statistical analyses to enhance a deeper understanding of how the results are obtained and the derived conclusions. Also, the manuscript mentions the positive correlation between cyclone activity and precipitation but does not delve into the causal mechanisms. These considerations, and the ones below, are important for a more compelling paper, and while they currently prevent me from giving my full endorsement for publication in its current state, I eagerly anticipate reviewing a revised version.

Response:

We would like to express our sincere gratitude for your help and supports. The most of your suggestions have been accepted and the manuscript is revised accordingly. Our responses to the comments are listed in a one-by-one manner as follows.

Abstract: The current abstract resembles a list of outcomes instead of a storyline, so it might benefit from a stronger overarching structure that seamlessly connects the climatological characteristics of Jianghuai cyclones, their correlation with precipitation in the Yangtze River region, and the atmospheric anomalies associated with their activity.

Response:

Thank you for your valuable suggestion. We have rewritten the relevant abstract, as follows: "**Abstract.** This study examines the climatic characteristics of 202 Jianghuai cyclones and their linkage with precipitation during the Meiyu period from 1961 to 2020. The results show that cyclones mainly originate from eastern Hubei Province and south-central Anhui Province, and further explored the statistical characteristics of the strength, radius, and their positive correlation. When studying the interdecadal variation of cyclones, we found that there is a similar trend between the interdecadal variation of cyclones and Meiyu precipitation. Therefore, we further investigate the correlation between the Jianghuai cyclones and the precipitation during the Meiyu period. There is a positive correlation coefficient of 0.769 between them. It's worth mentioning that the percentage

of precipitation affected by cyclone activities can reach up to 47%. The anomalous increase in precipitation caused by cyclones above 27°N can reach a maximum of 7 mm/day. When the cyclone existed, there was a significant altitude anomaly of negative geopotential height can be traced to day -4 at the 500 hPa level over Mongolia. The abnormally enhanced WPSH, southwest jet and negative geopotential height are the dominant factors causing abnormal precipitation during Jianghuai cyclones. Before and after the cyclone developed, water vapor flux and divergence from low latitudes abnormally increased. These provide sufficient water vapor conditions for the generation of cyclone precipitation."

Section 1: Please provide a more extensive description of the Meiyu front and period as they represent specific manifestations of the East Asian monsoon that may not be widely recognized worldwide.
Response:
     We appreciate your suggestion, and we have added more widely descriptions of Meiyu. In the revised manuscript, specifically in lines 27-39, we describe the adjustments we've implemented: "Meiyu is a special rainy season due to the progress of the East Asian summer monsoon. The East Asian summer monsoon broke out in the South China Sea in mid-May and then advanced northward, forming rain bands in South China, the Jianghuai region, the Korean Peninsula and Japan (Ding et al., 2004,2007; Qian et al., 2000). The name for this special rainy season is called Meiyu in China, while it is called Changma in South Korea and it is called Baiu in Japan (Ninomiya et al., 1987; Oh et al., 1997; Saito. 1995;). Meiyu front is one of the important weather systems affecting summer precipitation in the middle and lower reaches of the Yangtze River (Pang et al., 2013; Wang et al., 2014; Zhou et al., 2022; Tao et al., 1979). From mid-June to early July, the east of Yichang, Hubei Province, has continuous rains and short sunshine. These conditions are accompanied by heavy rainfall, strong wind and other weather phenomena in these areas during the Meiyu period (Ding. 1992; Zhao et al., 2021; Zhou et al., 2016)."

Adding a map to illustrate the location of Yangtze River, Jianghuai, and its surrounding provinces/regions would enhance the reader's understanding of the geographical context. It could also include the Meiyu front location. This visual aid would be valuable for situating the study area and providing context for readers who may not be familiar with the region.
Response:
     Thank you for your valuable suggestion. We added a map to illustrate the location of Yangtze River and Jianghuai cyclone. The picture is shown below:

[Figure]

Fig.1 Schematic diagram of the main weather system and the structure of temperature and pressure field in the middle and low levels of the Jianghuai cyclone. (Red dotted line: isotherm; Solid black line: contour line; Blue dot: precipitation area; Solid orange line: 500 hPa upper-level trough; Red arrow: low level jet; Black dotted line: warm inverted trough; Solid red line: warm shear; Solid blue line: cold shear; Letter C: cyclone; Letter A: WSPH.)

L.39. I think there might be a typo: floors -> floods?
Response:
    Thank you for the suggestion. We're sorry that we used the wrong word here. We have revised "floors" to "floods". The complete sentence in L.42 is "Historically, most of the summer floods disasters are caused by precipitation anomalies in the Meiyu period."

L.117. Please specify that the "National Meteorological Information Center" is part of the "China Meteorological Administration" to avoid confusions with similar institutions from other countries.
Response:
    We appreciate your suggestion, and we're sorry we didn't specify it clearly here. we have added "China Meteorological Administration" in L.131. The complete sentence is "The precipitation data are from the CN05.1 grid point observation dataset compiled by the National Meteorological Information Center of China Meteorological Administration with a resolution of 0.25°×0.25°."

Section 2.2: The description of tracking methods is presented at a technical level without a comparative analysis of their strengths and weaknesses. A more descriptive approach would offer insights into why the chosen method is ideal for this study, providing a more comprehensive methodological evaluation.
Response:
    Thank you for your valuable suggestion. We added some technical descriptions to explain why we chose this tracking methods in L.156-164. "Among them, the most commonly used cyclone tracking methods are the mean sea level pressure method (SLP) and 850 hPa relative vorticity method. Mailier et.al (2006) and Zhang et.al (2012) studied the tracks of individual cyclones in these two methods. Both of them found 850 hPa relative vorticity method can identify and detect

cyclone center earlier than the SLP method (Mailier et.al., 2006). The reason for this result is that SLP is easily affected by topography and large-scale background circulation shear vorticity (Hodges, 1994; Sinclair, 1994). So based on this advantage of the relative vorticity method, we select the 850 hPa relative vorticity tracking method."

L.154. It could be interesting to add a rectangle that illustrates this region, possibly incorporating it into the new figure.
Response:
    Thank you for your valuable suggestion. We add a rectangle to illustrate the region of 108°E-123°E, 28°N-35°N in Fig.1. The changed picture is shown below:

[Figure]

L. 165. When referring to the frequency of occurrence, what are the units? From the numbers, I would say that it refers to the total number of cyclones during the Meiyu period from 1961 to 2020. Please specify in the text, here and hereafter, and add to figures and/or figure captions as well.
Response:
    We appreciate your suggestion. The frequency of occurrence refers to the total number of cyclones during the Meiyu period from 1961 to 2020. We have specified in the L. 182 and after. "Figure 1b shows the frequency of cyclone occurrence refers to the total number of cyclones during the Meiyu period from 1961 to 2020. The genesis locations of cyclones are mainly located in the middle and lower reaches of the Yangtze River and the Huaihe River basin, with an east–west band distribution (Wang et al., 2015; Wu et al., 2020). The frequency of occurrence refers to the total number of cyclones during the Meiyu period from 1961 to 2020 is higher in the region of the Hubei and Chongqing junction, eastern Hubei, northern Jiangxi, south-central Anhui, Jiangsu and Zhejiang. Research has found that the genesis locations of cyclones are closely related to the landform (Xu 2021; Zhang et al., 2012)."

L.245. Please add the definition of "Meiyu intensity index" as it is not defined in the text. It is important to know how this index is defined to be able to fully follow and understand the discussions

where it is used.

Response:

Thank you for the suggestion. We have added the definition of "Meiyu intensity index" in L.272-L.283. we describe the adjustments we've implemented: "The Meiyu intensity index is defined as:

$$M = \frac{L}{L_0} + \frac{0.5(R/L)}{R_0/L_0} + \frac{R}{R_0} - 2.5$$

M is the Meiyu intensity index. L is the length of the Meiyu in a given year (unit: day) and $L_0$ means the average length of the Meiyu over the years (units: day). R is the total precipitation of Jianghuai River basin during Meiyu in a given year, and $R_0$ is the average total precipitation of Jianghuai River basin during Meiyu over the years. Where M between -0.375 and 0.375, China Meteorological Administration defines this year as the normal. Where M between 0.375 and1.25, this year is defined as a little strong. Where M greater than or equal to 1.25, this year is defined as strong. Where M between -1.25 and -0.375, this year is defined as a little weak. Where M less than or equal to -1.25, this year is defined as weak (GB/T 33671-2017)."

L.225, and so on. The term "anomaly/anomalies" is used throughout the text without explicitly defining it. I guess it takes the whole Meiyu period from 1961 to 2020 as the base state; however, it is not defined in the text. Please do.

Response:

We appreciate your suggestion, and we have added the definition about "anomaly/ anomalies" in L.320-L.324. "Figure 7c shows the spatial distribution of the daily mean precipitation anomaly of the Jianghuai cyclone. The shaded part indicates that the 95% confidence interval is passed according to the T test. The anomaly is based on the whole Meiyu period from 1961 to 2020 (The exceptions mentioned below are also based on the whole Meiyu period from 1961 to 2020)."

L. 334 -344. Are these results from the present study and backed up by the references provided or are the results of the references? Please clarify.

Response:

We're sorry that the references here have given you the wrong impression. These results are from the present study and backed up by the references provided. We adjust the description of result in L.361-380. "Therefore, the abnormal precipitation caused by the Jianghuai cyclone mainly comes from the abnormal southwest winds and the strengthening of the WPSH. The enhanced southwest jet provides sufficient warm and moist air for the formation of cyclones and promotes the eastward migration of cyclones after formation. Liu et al. (2020) and Zhao et al. (2021) studied the causes of the super strong Meiyu year in 2020, mentioned that the WPSH is unusually strong and westward accompanied by an abnormal increase in precipitation. Liu et al. (2020) found that the enhanced southwest jet stream is conducive to the development of vertical movement in the middle and low levels, which provides the necessary dynamic conditions for the formation of sustained precipitation during the Meiyu in 2020.

Cold air activity is one of the important factors for the formation of heavy precipitation, which can promote the convergence and uplift of low level necessary for heavy precipitation (Liu et al.,

2020). The enhanced negative geopotential anomaly over Mongolia provides cold and dry air brought by the westerly jet for cyclone development. The increasing frequency of cyclones over the Yangtze River and Huaihe River leads to the abnormal increase in precipitation in the middle and lower reaches of the Yangtze River during the Meiyu period. However, due to the strengthening of the WPSH, the southern part of China is controlled by the abnormal positive geopotential height, and the precipitation decreases. Zhao et al. (2021) also found that when the WPSH enhanced, there was a decrease in precipitation in South China."

Section 4: It is advisable to briefly mention potential directions for future research, identifying unexplored questions and areas for further study to build upon the presented findings. For example, the specific mechanism of the southwest jet's influence could be further analyzed and researched.
Response:

Thank you for your valuable suggestion. We have added potential directions for future research, identifying unexplored questions and areas for further study to build upon the presented findings in L.490-505. "We explored the cyclone characteristics and study emphasizes the link between cyclone activity and Yangtze River precipitation. Spatially, abnormal precipitation patterns are identified, tracing the evolution of geopotential height anomalies and water vapor flux. But the specific mechanism by which the southwest jet affects cyclones during the Meiyu period is not clear enough. Zhang et al. (2018) suggest that the strengthening of the Southwest jet will lead to the development of α mesoscale low-pressure disturbance near the Meiyu Front and the occurrence of extreme precipitation. Liu et al. (2020) found that the strengthening of the southwest jet made the southerly meridional strong gradient zone on the north side of the meridional wind maximum center move northward in the low-level dynamic conditions of the rainstorm process during Meiyu. How the Southwest jet stream influences the development of physical factors to promote the formation of Jianghuai cyclones remains to be considered and analyzed. Zhao et al. (2010) found that the causes of Jianghuai cyclones with different intensities were different through a case study. Therefore, we think it is also necessary to consider the difference in the influence of different intensities of Jianghuai cyclones on precipitation. These problems need further analysis and research."

Fig. 1: It would be beneficial to have the same size in both subfigures. In the caption, I think the parentheses containing two sentences should be integrated as part of the text.
Response:

We appreciate your suggestion, and we have adjusted the layout of pictures. The changed picture is shown below:

[Figure]

Fig 1. Distribution of the cyclone genesis locations, tracks (a) and the frequency of genesis locations refers to the total number of cyclones (b) during the Meiyu period from 1961 to 2020 (The brown dots represent the genesis locations. The yellow lines indicate the tracks).

Fig. 2: I would change the y-labels to "Number of cyclones" for easier interpretation at first glance. Additionally, the units of intensity are hard to read in the figures. Consider enlarging the figures or adjusting their layout to improve visibility.

Response:

Thank you for your valuable suggestion. We have changed the y-labels to "Number of cyclones" and enlarged the figures or adjusted their layout in Fig.2. The changed picture is shown below:

[Figure]

Fig 2. Distributions of the number of selected cyclones versus their (a) intensities (units: $10^{-5}$ s$^{-1}$), (b) radii (units: km), and (c) lifetimes (units: days).

Fig. 3: I don't understand what "Point" in figure b means. Please clarify or correct.
Response:

    We're sorry that the references here have given you the wrong understanding. It means the difference value of track step between the maximum intensity and the radius of the cyclone. We have adjusted in the picture and text.

    "From the distribution of difference value of track step between the maximum intensity and the radius of the cyclone shown in Figure 3b:

[Figure]

Fig 3. Correlation between maximum intensity (units: $10^{-5}$ s$^{-1}$) and maximum radius (units: km) (a) and their difference value of track step during the development of the Jianghuai cyclone in the Meiyu period (b).

Fig. 4, 6: I think "Number" in the y-labels refers to the "number of cyclones per year". Please add and specify both in the caption and the text.
Response:
  We appreciate your suggestion, and we have changed the y-labels to "Number of per cyclones" in Fig.4,6.

[Figure]

Fig 4. Periodic wavelet analysis diagram of Jianghuai cyclones during the Meiyu period from 1961 to 2020 (units: number of cyclones per year) (shadow indicates passing the 95% confidence interval according to the T test).

[Figure]

Fig 6. (a) Changes in precipitation (blue bar chart) (units: mm/day) and the number of cyclones per year (orange line); (b) intensity index (blue line) and the number of cyclones per year (orange line) in the Meiyu period from 1961 to 2020.

Fig. 7 and 8: Again, consider enlarging the figures or adjusting their layout to improve visibility.
Response:
    Thank you for your valuable suggestion. We have enlarged the figures or adjusted their layout in Fig.7. The changed picture is shown below:

[Figure]

Fig 7. (a) Annual mean precipitation during the Meiyu period from 1961 to 2020 (units: mm/year); (b) proportion of Jianghuai cyclone precipitation relative to total precipitation during the Meiyu period (units: %); (c) daily mean precipitation anomaly of the Jianghuai cyclone during the Meiyu period (units: mm/day) (shadow indicates passing the 95% confidence interval according to the T test).

[Figure]

Fig 8. Evolution of composite geopotential height anomalies (shading; units: gpm) and horizontal wind anomalies (units: m/s) on the 850 hPa, 700 hPa, and 500 hPa isobaric surfaces for day −4 (a–c), day −2 (d–f), day 0 (g–i) and day +2 (j–l) for the 202 selected Jianghuai cyclones. Shading

indicates that composite geopotential height anomalies are significant at the 95% confidence level based on a T test. Vectors are plotted if wind anomalies are significant at the 95% confidence level based on a T test in at least one direction.

Also, please consider relocating the "T test" specification to the main text when discussing confidence levels, rather than including it in the figure captions. Include it the first time it is mentioned and subsequent times if you deemed necessary or use the hereafter expression.
Response:
     Thank you for your valuable suggestion. We have relocated the "T test" specification to the main text when discussing confidence levels. The modified content is as follows:

L.230-232: "The frequency of Jianghuai cyclone occurrence refers to the total number of cyclones is characterized by multiperiod variation (Figure 5). The shaded area in the figure indicates that the 95% confidence interval according to the T test is passed.
Please, review the text and figure captions to explicitly specify the units of the magnitudes."

L.320-322: "Figure 7c shows the spatial distribution of the daily mean precipitation anomaly of the Jianghuai cyclone. The shaded part indicates that the 95% confidence interval is passed according to the T test."

L.339-342: "Figure 8 shows the evolution of composite geopotential height and horizontal wind anomalies for three different levels of Jianghuai cyclones from day -4 to +2 during the Meiyu period. Composite geopotential height anomalies are significant at the 95% confidence level based on a T test."

L.410-413: "Figure 10 shows the distribution of water vapor flux anomalies and water vapor flux divergence anomalies at the pressure level of 850 hPa during the Jianghuai cyclone from day -2 to day +2. The color field and wind vector arrows in the figure both passed the 95% significance according to the T test."

---

## Author Comment (AC6)

* * *
**Responses to reviewers' comments**

**In this response file, the text in black shows the comments from reviewers and editor, while the text in blue is our replies.**
* * *
The authors employed the relative vorticity method to track Jianghuai cyclones and subsequently investigated their climatological characteristics, including frequency, intensity, and radius, in relation to Meiyu precipitation. This study holds significant value and serves as a crucial foundation for further research on the dynamics of Jianghuai cyclones and their impact on both mean and extreme precipitation. However, the current study requires substantial revisions. The key strength of this research lies in the association between Jianghuai cyclones and Meiyu precipitation; however, the analysis in this aspect is relatively limited. Therefore, I recommend that the authors devote more attention to this particular area. For instance, they could explore the differences in circulation patterns and underlying mechanisms between cyclone precipitation days and non-cyclone precipitation days, or investigate the connection with extreme precipitation events.

Response:

We would like to express our sincere gratitude for your help and supports. The most of your suggestions have been accepted and the manuscript is revised accordingly. Our responses to the comments are listed in a one-by-one manner as follows.

Detailed Comments:

Line 116: CN05.1 data need a citation.

Response:

We are very sorry that we forgotten the data reference, and we've added citations to the references in Line 131. "The precipitation data are from the CN05.1 grid point observation dataset compiled by the National Meteorological Information Center of China Meteorological Administration with a resolution of 0.25°×0.25° (Wu et al., 2013; Xu et al., 2009). "

Section 2.2 Methods:

I would suggest the authors giving a briefly introduction of the rationale of the vorticity tracking method proposed by Hodges firstly, and then the details would be more readable. Besides, the method (Hodges, 1994, 1995) introduction still left too much attention to detail and seems tedious. And the advantages of the method used in this study over other methods should be stressed. That is, the last two paragraphs in this section could be rewritten to improve readability.

Response:

We appreciate your suggestion, and we have rewritten the last two paragraphs. In the revised manuscript, specifically in lines 147-169, we describe the adjustments we've implemented: "Scholars have proposed a number of methods to identify extratropical cyclones. The objective identification and tracking method for cyclones used in this paper is the vorticity tracking method proposed by Hodges (1994, 1995). This method mainly uses the relative vorticity field at the 850

hPa to determine the feature points of the cyclone. Feature points are used to correspond to the position of the cyclone and to match the cyclone track within a given time span. In addition to the relative vorticity method of tracking proposed by Hodges, different methods of cyclone identification have also been proposed by other scholars. Lu (2017) improved the extratropical cyclone identification and tracking method involving the nine-point pressure minimum. Jiang et al. (2020) proposed an algorithm for identifying extratropical cyclones on the basis of gridded data. This algorithm is named the eight-section slope detection method.

Among them, the most commonly used cyclone tracking methods are the mean sea level pressure method (SLP) and 850 hPa relative vorticity method. Mailier et.al (2006) and Zhang et.al (2012) studied the tracks of individual cyclones in these two methods. Both of them found 850 hPa relative vorticity method can identify and detect cyclone center earlier than the SLP method (Mailier et al., 2006). The reason for this result is that SLP is easily affected by topography and large-scale background circulation shear vorticity (Hodges, 1994). So based on this advantage of the relative vorticity method, we select the 850 hPa relative vorticity tracking method. The relative vorticity tracking method can detect low vortex systems earlier and track cyclones for a longer period of time with better stability. When the closed pressure levels are not visible on the satellite map, the vorticity tracking method can still continue to track the cyclone, improving the accuracy of cyclone track data."

Line 155-156: "genesis location" is the repetition of "the first occurrence". I would suggest changing to be: The brown dots represent the genesis locations, i.e., the first place meeting the criterion, of the Jianghuai cyclone.
Response:
    Thank you for your valuable suggestion. We have changed the the description of "the first occurrence" to "The brown dots represent the genesis locations, the first place meeting the criterion, of the Jianghuai cyclone." In the Line 178-179.

Lines 157-161: The authors have pointed out that the tracks of the cyclone can be categorized into two group, the easterly and the northeasterly. However, since there are no further discussions for the two groups respectively. Indeed, both the easterly or northeasterly paths are related to the locations of the WPSH.
Response:
    We appreciate your suggestion. We have revised the relevant description in line 180-182: "As shown in the figure, most of the cyclones develop in the Jianghan Plain and southern Anhui Province, then move eastward to the Yellow Sea coast. Some cyclones move northward through Shandong Province and reach the Bohai Sea."

Line 165-166: Two centers with high values, i.e., southwestern Hubei and eastern Hubei. The original sentences is puzzling, and I suggest the authors revising it.
Response:
    Thank you for your valuable suggestion. Our revised content is as follows in line 189-191: "The frequency of occurrence refers to the total number of cyclones during the Meiyu period from 1961 to 2020 is higher in the region of the Western Hubei Province and Eastern Hubei Province."

1: "(The brown dots represent the genesis locations. The yellow lines indicate the tracks)." The bracket is no need.

Response:

We appreciate your suggestion, and we have deleted the bracket in the article.

Line 176: ". The larger the relative vorticity intensity is, the stronger the cyclone intensity is." seems redundant.

Response:

Thank you for your valuable suggestion. We have removed relevant redundant descriptions such as these from the original text.

Line 178: It is confusing to see a "$0\times10-5$ s $-1$" in the relative vorticity for the Jianghuai cyclone.

Response:

We appreciate your suggestion. We have revised the relevant description in line 200-201: "Figure 3a shows that among the 202 selected cyclones, the intensity of the cyclone center mainly ranges from $1.5\times10-5$ s-1 to $7.3\times10-5$ s-1."

Line 186: "radii" to "radius"?

Response:

Thank you very much for your question. "radii" is the plural form of "radius".

Line 187: "time" to "time span"? This is different from the caption of Fig. 2, in which the "time" is the "lifetime"? Which is right? The description in the main text or the figure caption of Fig. 2?

Response:

We appreciate your suggestion, and we're sorry we didn't specify it clearly here. After the revision, we added an explanation about "lifetime" in line 199. "The lifetime is defined as the time of cyclones affecting precipitation on land."

Lines 193-197: Several places in the manuscript are repetitive. I suggest the authors revised them carefully. Below is an example. "Figure 3a shows a positive correlation between the maximum intensity and the maximum radius of cyclone development. The stronger the intensity of a cyclone is, the larger its radius. Therefore, the horizontal scale of most strong cyclones is larger than that of weak cyclones, the precipitation is greater, and the precipitation range is larger."

Response:

We appreciate your suggestion, and More tautology has been removed. The revised statement as follows in Line 201: "Figure 3a shows that among the 202 selected cyclones, the intensity of the cyclone center mainly ranges from $1.5\times10-5$ s-1 to $7.3\times10-5$ s-1. The number of cyclones in the range of $2\times10-5$ s-1 to $3\times10-5$ s-1 has the largest proportion, accounting for 36% of the total number of cyclones. A total of 180 cyclones are in the range of $1.5\times10-5$ s-1 to $5\times10-5$ s-1 in intensity, accounting for 89%. Figure 3b shows the relationship between the radius of cyclones and the number of cyclones. Most of the cyclones have an average radius between 300 and 800 km, accounting for 96% of the total number. The number of cyclones with radii between 500 and 600 km is the largest, accounting for 35%. Figure 3c shows the relationship between the time of cyclones affecting precipitation on land and the number of cyclones. Most of the cyclones affect precipitation

on land for 1-3 days, and only one cyclone affects precipitation on land for more than 3 days. The number of cyclones' lifetime that affected precipitation on land within 2 days was 186, accounting for 92% of the total number."

Line215: "Figure 4a shows a positive correlation between the maximum intensity and the maximum radius of cyclone development. Therefore, the horizontal scale of most strong cyclones is larger than that of weak cyclones, the precipitation is greater, and the precipitation range is larger."

Line 215: Indian Ocean dipole (IOP) to "Indian Ocean Dipole (IOD)"?

Response:

We are very sorry about this mistake, and we have revised it in line 236-244. "This quasiperiodic variation component is mainly influenced by the out-of-ocean forcing of the Indian Ocean dipole (IOD), which changes from the ENSO in the previous winter to late spring and early summer with seasonal changes (Liang et al., 2018). During the positive phase of the IOD, the strong warming of the Indian Ocean triggers a strong Indian monsoon. This leads to a strengthening of the WPSH and an increase in precipitation in southern China. The southwesterly low-level jet, which are enhanced by the positive IOD, also provide sufficient water vapor and warm advection to generate favorable conditions for the development of the Jianghuai cyclone."

Line 219: "Southwestern rapids" to "Southwesterly low-level jet"?

Response:

We are very sorry that there were some lexical errors and we have corrected them in lin242-243: "The southwesterly low-level jet, which are enhanced by the positive IOD, also provide sufficient water vapor and warm advection to generate favorable conditions for the development of the Jianghuai cyclone."

Lines 228-230: The alternatively increase and decrease of the numbers of Jianghuai cyclones is indeed the "decadal variation of Jianghuai cylone", with positive anomaly in 1965-1970, 1990-2000, and 2000-after, and negative anomaly in 1970-1990 and 2000-2010.

Response:

We appreciate your suggestion. We have revised the relevant description in line 252-254: "In the long term, the frequency of cyclone activity in the middle and lower reaches of the Yangtze River with positive anomaly in 1965-1970, 1990-2000, and 2000-after, and negative anomaly in 1970-1990 and 2000-2010."

Line 230-232: I would suggest the authors illustrating the relationship between the decadal variation of Jianghuai cyclone and that of Meiyu precipitation.

Response:

Thank you for your valuable suggestion. We have revised the relevant description in line 2566-258: "The decadal variation of precipitation during the Meiyu period with positive anomaly in 1965-1970, 1995-2000 and 2010-after, and negative anomaly in 1970-1980, 1985-1995 and 2000-2010."

Line 242: I would suggest removing "gales" since there is no further discussion.

Response:

We appreciate your suggestion, and we have removed "gales" in line 268: "They form rainstorms in the middle and lower reaches of the Yangtze River and the coastal areas."

Line 245: "Meiyu intensity index" should be defined explicitly.

Response:

Thank you for your valuable suggestion. We have defined "Meiyu intensity index" in line 133-146. "We used the Meiyu intensity index to characterize the strength of Meiyu, and data is from the National Climate Center of China. The area for which the Meiyu intensity index is calculated is defined in the article (GB/T 33671-2017). Meiyu intensity index is defined as:

$$M = \frac{L}{L_0} + \frac{0.5(R/L)}{R_0/L_0} + \frac{R}{R_0} - 2.5$$

M is the Meiyu intensity index. L is the length of the Meiyu in a given year (unit: day) and $L_0$ means the average length of the Meiyu over the years (units: day). R is the total precipitation of Jianghuai River basin during Meiyu in a given year, and $R_0$ is the average total precipitation of Jianghuai River basin during Meiyu over the years. Where M between -0.375 and 0.375, China Meteorological Administration defines this year as the normal. Where M between 0.375 and 1.25, this year is defined as a little strong. Where M greater than or equal to 1.25, this year is defined as strong. Where M between -1.25 and -0.375, this year is defined as a little weak. Where M less than or equal to -1.25, this year is defined as weak."

Lines 250 and Lines 252: "0.769" and "0.760" to be "0.77" and "0.76".

Response:

We appreciate your suggestion, and we have revised the relevant description in line 277-281: "We found that the number of cyclones has a positive correlation coefficient of 0.77 with precipitation in the Meiyu period passing the 99% confidence interval according to the student's t-test. The number of cyclones was also positively correlated with the Meiyu intensity index, with a correlation index of 0.76 passing the 99% confidence interval according to the student's t-test."

Line 259: "annual average" to "annual total"?

Response:

We are very sorry that there were some lexical errors and we have corrected them in line 2285-286: "Figure 8a shows the spatial distribution of mean annual total precipitation during the Meiyu period from 1961 to 2020."

Line 267-269: The definition of non-cyclone precipitation days should be explicitly stated.

Response:

We are very sorry that the explanation here gives rise to a misunderstanding. We have removed this definition.

Lin 387: The colored region passed the 95% confidence interval according to a test. Student's t-test?

Response:

We appreciate your suggestion, and we have changed all "T test" to "Student's t-test". The revised text will not be shown here.